# Porcine Epidemic Diarrhea: Insights and Progress on Vaccines

**DOI:** 10.3390/vaccines12020212

**Published:** 2024-02-18

**Authors:** Jung-Eun Park

**Affiliations:** Laboratory of Veterinary Public Health, College of Veterinary Medicine, Chungnam National University, Daejeon 34134, Republic of Korea; jepark@cnu.ac.kr

**Keywords:** porcine epidemic diarrhea, coronavirus, vaccine

## Abstract

Porcine epidemic diarrhea (PED) is a swine-wasting disease caused by coronavirus infection. It causes great economic damage to the swine industry worldwide. Despite the continued use of vaccines, PED outbreaks continue, highlighting the need to review the effectiveness of current vaccines and develop additional vaccines based on new platforms. Here, we review existing vaccine technologies for preventing PED and highlight promising technologies that may help control PED virus in the future.

## 1. Introduction

Porcine epidemic diarrhea (PED) is a gastrointestinal illness in swine that is triggered by the porcine epidemic diarrhea virus (PEDV). This virus can infect pigs of any age; however, it is particularly lethal and causes high rates of illness in newborn piglets. The primary method of PED transmission is through fecal-to-oral contact, although it can also spread via airborne particles from the feces to the nose [1]. PEDV predominantly targets the intestinal epithelial cells of pigs, leading to degeneration, necrosis, and loss of intestinal villi. Infection of these cells results in impaired nutrient absorption and triggers symptoms such as vomiting, diarrhea, weight loss, appetite loss, and potentially death [2,3].

PED was initially identified in the United Kingdom in 1971. Following identification, the disease quickly spread across various European nations, leading to substantial mortality among suckling piglets in Europe throughout the 1970s and 1980s [4]. In Asia, the first cases of PED emerged in the early 1980s, and the disease has been endemic in several Asian countries [5,6,7]. Until 2010, the spread of PED was primarily confined to Europe and Asia, and it was not regarded as a worldwide concern. However, the situation changed dramatically in early 2013 when the PED unexpectedly emerged in the United States [8,9,10] and rapidly spread to neighboring countries in North America [11,12,13,14]. This outbreak eventually reached East Asia [15,16,17] and Europe [18], culminating in a global PED epidemic during 2013–2014.

PEDV was first discovered in 1978 by scientists from Belgium [19,20]. A member of the *Alphacoronavirus* genus within the *Coronaviridae* family, PEDV is part of the *Nidovirales* order. The virus is encased in a nearly spherical shell measuring between 95 and 190 nanometers in diameter and features a club-like trimeric spike that is approximately 18 to 23 nanometers long [19]. Its genetic material is composed of a single-strand, positive-sense RNA approximately 28 kilobases in length with a 5′-cap and a 3′-polyadenylated tail at its ends. There are seven key coronavirus genes within the PEDV genome, including open reading frame (ORF) 3, which contains the following components: 5′ untranslated region (UTR)-ORF1a-ORF1b-S-ORF3-E-M-N-3′ UTR [21,22]. The ORF1a and ORF1b segments are precursors to polyproteins that are split into 16 nonstructural proteins (nsp1–nsp16) by the virus’s protease, aiding in virus multiplication and evasion of the host’s immune defenses. The S glycoprotein can form trimeric spikes on the viral surface, which mediate cell entry by attaching to host receptors and facilitating membrane fusion. The S gene, known for its high variability among PEDV strains, is used for phylogenetic analysis to determine viral genetic diversity [23,24,25]. PEDV is classified into two main genogroups: genogroup 1 (G1) and genogroup 2 (G2) [26]. G1 is further divided into the original G1a and the recombinant G1b, while G2 is split into the locally epidemic G2a, the globally epidemic G2b, and the recombinant G2c [22]. The G1a genogroup includes the CV777 strain, the original strain identified in Belgium, along with genetically similar strains. G2 encompasses recent field isolates divided into G2a, linked to past and present regional outbreaks in Asia, and G2b, associated with the 2013–2014 epidemic and major strains currently found in the United States and Asia. The evolution of PEDV has been complex and rapid, with potential recombination events occurring between the G1a and G2 lineages and leading to new lineages such as S-INDEL (G1b) and the newly identified G2c [10,27,28,29]. Changes in the S gene through mutation and recombination can alter the pathogenicity of the virus and its ability to infect different tissues or species [30,31,32,33,34,35,36,37]. The S protein is also vital for infection, as it induces the production of neutralizing antibodies and is a key target for vaccine development, both for live attenuated and subunit vaccines [22]. The E protein, the virus’s smallest structural component, plays a significant role in virus assembly and release [38], while the M protein is essential for the formation of the coronavirus envelope [39]. The presence of both the E and M proteins is sufficient to create virus-like particles (VLPs) [40,41]. The M protein, found throughout the cytoplasm, can also induce cell cycle arrest in the S phase via the cyclin A pathway. The N protein, a highly conserved phosphoprotein, exhibits only minor mutations across different strains and is involved in several viral life cycle processes, such as viral RNA synthesis regulation, viral RNA packaging into helical nucleocapsids, and virus assembly [42,43].

While herd immunity and stringent biosecurity measures represent the foremost strategies for preventing PEDV infection, the continual appearance of novel variants, particularly those originating from recombination events, has resulted in vaccine inefficacy, posing significant challenges to preventing and controlling PED. This article examines the present situation regarding vaccination efforts aimed at eradicating PEDV.

## 2. Vaccination Strategy for PEDV

Piglets are born without immunoglobulins (agammaglobulinemic) and remain immunodeficient until weaning due to the restriction of maternal immunoglobulin transfer by the pig placenta [44]. As a result, the primary defense of newborn piglets is mediated through lactogenic immunity. During lactation, immunoglobulins such as secretory IgA (sIgA), IgG, and IgM are passively transferred from sows to piglets via colostrum and milk [44,45,46,47]. Colostrum is rich in IgG, which is derived from the sow’s serum and is absorbed by the piglet in the first 24 to 48 h after birth. After the transition from colostrum to milk, at approximately 3 to 4 days, sIgA becomes prevalent [48]. Antibody-secreting cells in the mammary gland, which migrate from the intestine at the end of pregnancy, ultimately produce sIgA [49].

Insights from the development of a transmissible gastroenteritis virus (TGEV) vaccine, another catastrophic enteric coronavirus, laid the groundwork for lactogenic immunity and vaccination tactics for such diseases [45,47,50,51,52]. Studies have highlighted the correlation between piglet protection and high sIgA levels in milk but not with IgG levels in serum or colostrum. Sows with natural TGEV infection or oral inoculation produced sIgA, whereas those immunized with inactivated TGEV mainly produced IgG [45,48,51,52]. Similarly, sows parenterally vaccinated with PEDV developed a specific immune response but failed to protect their piglets fully [53,54]. The degree of viral replication in the sow’s gut influences the induction of adequate production of IgA and neutralizing antibodies in milk and colostrum [48]. Therefore, the vaccine administration route is crucial for inducing mucosal immunity in sows. To protect newborn piglets from enteric coronaviruses such as PEDV, passive lactogenic immunity via the gut-mammary-sIgA axis remains the most effective strategy [48]. Oral vaccination of sows seems to be the optimal method for enhancing maternal immunity levels. Although most enteric coronavirus vaccines are designed to induce lactogenic immunity via the vaccination of sows, they are often administered systemically. Sows given orally attenuated TGEV showed lower milk sIgA antibody levels and piglet protection than did those inoculated with virulent TGEV strains, likely due to the attenuated vaccine’s lower viral dose, diminished gastrointestinal stability, and reduced viral replication in the sow intestines.

## 3. Current Status of PED Vaccines

To manage PEDV outbreaks, a variety of vaccines have been formulated (Table 1). Traditional vaccines against PEDV predominantly consist of whole-virus forms, including inactivated and attenuated versions. However, research has been conducted on other forms, such as subunit, nucleic acid, viral vector, and VLP vaccines. The advantages and disadvantages of each type of vaccine platform are detailed in this review.

### 3.1. Live Attenuated Vaccine

Live attenuated vaccines (LAVs) are renowned for their high immunogenicity and often require only a single dose to establish protective immunity. The traditional method of creating LAVs involves serial passaging through non-natural host tissue cultures, which leads to attenuation.

In 1994, the PEDV CV777 strain, which was attenuated after 28 passages in cell culture, was shown to protect more than 80% of piglets post-immunization [55]. Tong et al. reported the production of an LAV developed through in vitro serial passaging of CV777, which exhibited a 95.52% protection rate and a 96.2% passive immunization protection rate in piglets aged three to six days [56]. In 1999, a bivalent LAV for PEDV and TGEV was developed, which provided active and passive protection rates against PEDV of 97.7% and 98%, respectively [57,58]. These vaccines were extensively used in China, effectively controlling PEDV and TGEV spread until 2010, when highly virulent PEDV variants emerged. In 2015, China approved a trivalent LAV (PEDV, TGEV, and rotavirus). This vaccine included the highly virulent pandemic virus strain CT, which, after 120 passages in Vero cells (P120), exhibited higher viral titers, improved cell adaptability, and more pronounced cytopathic effects. Vaccination with P120 significantly reduced clinical symptoms in piglets, resulting in a 100% survival rate [2]. Japan and South Korea have also developed vaccines. The Japanese PEDV strain 83P-5, which was attenuated through serial passages in Vero cells, is available as an LAV [59]. Inoculation of sows with 83P-5 conferred passive protection to 80% of piglets against the G2 PEDV challenge [60]. In South Korea, the virulent strains SM98-1 and DR-13 were attenuated through in vitro passaging. The SM98-1 strain is used either as an intramuscular LAV or an inactivated vaccine, while DR-13 is available as an oral LAV [61,62]. Song et al. demonstrated that the oral administration of DR-13 to late-term pregnant sows passively protected 87% of suckling piglets after a homologous challenge [62]. To date, several cell-attenuated G2 strains have been documented. These include the US isolate PC22A and the Asian strains YN, Pingtung-52, and KNU-141112 [63,64,65,66]. Hou et al. extensively reviewed the mutation patterns and molecular mechanisms associated with these four attenuated strains [67]. Generally, these cell-adapted strains exhibit attenuation in piglets while maintaining high immunogenicity, characterized by the elicitation of a high level of neutralizing antibodies.

While serial passaging can reduce the virulence of PEDV strains, it may also diminish their ability to confer resistance to challenges with parent strains. For instance, in a study of the cell culture-adapted PC22A strain, the virus at the 100th (P100) and 120th passages (P120) was fully attenuated in weaned pigs but only partially attenuated in neonatal piglets. Interestingly, compared with P120, P100 induced the production of greater serum PEDV IgA, IgG, and viral neutralization (VN) antibody levels and a greater number of PEDV IgA antibody-secreting cells post-challenge with the virulent strain [68]. This finding highlights the double-edged nature of PEDV attenuation. Fully attenuated LAV candidates in piglets might not stimulate sufficient lactogenic immunity in sows. Older pigs exhibit greater resistance to PEDV infection and disease than piglets [69]. Consequently, a fully attenuated virus that is appropriate for use in piglets can lead to inefficient replication and reduced viral immunogenicity in older pigs, resulting in inadequate protective immunity.

In response to this challenge, researchers have started to employ reverse genetics systems to modify PEDV strains, aiming for reduced virulence but high immunogenicity. A mutation in which the histidine (H) at position 226 in PEDV EndoU was replaced with alanine (A) led to early and strong activation of type I and III interferon (IFN) responses and significantly decreased viral virulence [70]. Deleting specific motifs in the C-terminus of the S protein and a portion of the N-terminal domain in the S protein reduced the pathogenicity of the virulent PEDV icPC22A, but the protective efficacy of this recombinant strain needs further examination [31,67]. Two recombinant PEDVs were created by mutating specific residues in nsp16 and the S protein, leading to the elimination of 2′-O-methyltransferase activity and abolishment of the S protein endocytosis signal. The KDKE4A and KDKE4A-SYA strains exhibited decreased replication efficiency but induced stronger IFN responses in vitro. The virulence of these strains was significantly reduced, with KDKE4A-SYA providing 80% protection in piglets [71]. Furthermore, variants containing mutations in nsp1, nsp2, nsp3, nsp5, nsp14, and the E protein are also potential candidates for PEDV LAVs. Mutations in these genes, known to significantly reduce viral virulence in other coronaviruses, such as murine hepatitis virus and severe acute respiratory syndrome coronavirus (SARS-CoV), are promising targets [72,73,74,75,76,77,78,79]. In summary, introducing mutations in virulence genes and deactivating IFN antagonists via reverse genetics offers a promising strategy for developing effective PEDV LAVs.

The efficacy of protection is crucial, but safety concerns significantly impede the use of PEDV LAVs. The attenuated strains, created either by serial passaging or molecular engineering, are designed to be nonpathogenic due to the introduction of mutations. However, these vaccine strains may revert to virulent strains, primarily through two mechanisms: (i) accumulation of mutations within the viral genome and (ii) recombination. There are two primary methods for producing promising attenuated vaccine candidates: (i) the classical approach of serially passaging viruses in non-natural hosts or environments, leading to adaptation to these new conditions and reduced virulence in the natural host, and (ii) the reverse genetics approach, which involves genetically modifying various genes to decrease pathogenicity while maintaining viral viability. Despite these strategies, the risk of virulence reversion exists owing to compensatory mutations elsewhere in the genome, particularly in immunocompromised hosts or when virus proliferation is less vigorous. The 2′-O-methyltransferase function of nsp16, a feature conserved among coronaviruses, is essential for the capping of viral genomic RNA during replication and transcription. A SARS-CoV nsp16 mutant (dNSP16) showed potential as a vaccine candidate in aged mouse models. However, in immunocompromised RAG^−/−^ mice, inoculation with the dNSP16 mutant led to weight loss and lethality in 62.5% (5/8) of the mice, indicating reversion to virulence [80]. Interestingly, while the targeted mutation in nsp16 remained, six additional mutations were identified in nsp3, nsp12, and nsp15, possibly acting as compensatory mutations. A similar situation was observed with the PEDV nsp14-ExoN mutant E191A, which showed significant attenuation but high genetic instability, with back mutations occurring both in vitro and in vivo [81]. To combat mutation-driven virulence reversion in coronavirus LAVs, a strategy involving multiple mutations to attenuate the virus through different pathways is suggested. Additionally, recombination is a key evolutionary factor for many RNA viruses, particularly coronaviruses [82], which can exhibit recombination rates of up to 20% during mixed infections with closely related strains [83]. Recombination center reversion, in which vaccine strains recombine with field virulent strains to create new variants, poses a significant challenge to LAV application. This phenomenon has been observed for other coronaviruses, such as infectious bronchitis virus [84] and PEDV [85]. In China, recombinant PEDV strains have been identified in several major pig-farming regions [85,86,87], with one strain exhibiting high pathogenicity in the field due to recombination between a low-pathogenicity vaccine strain and a virulent field strain [85].

Overall, the development of LAVs is constrained by the risk of reversion to their virulent wild-type forms [88,89]. Key considerations in developing an effective LAV include selecting strains that exhibit low virulence, high titers, and strong immunogenicity. Isolating PEDV strains in vitro, particularly the pandemic variable strains, poses significant challenges. Even successful isolation does not guarantee high viral titers.

### 3.2. Inactivated Vaccines

Inactivated vaccines are known for their exceptional safety and straightforward production process. However, their immunogenicity may be compromised during the inactivation procedure, often necessitating repeated and booster doses.

As early as 1993, studies indicated that a tissue-inactivated PEDV vaccine could confer immunity to piglets for up to six months [90]. In 1994, Ma et al. developed an inactivated vaccine using the cell-adapted CV777 strain, which was shown to provide 85.19% protection in 3-day-old pigs and an 85.0% passive immunization protection rate in piglets born to vaccinated sows [55]. The same team also formulated a bivalent inactivated vaccine for TGEV and PEDV, which became commercially available in China in 1995 [91]. However, due to the inability of inactivated vaccines to replicate within the host, they require multiple administrations and high immunization doses. In 2016, an inactivated vaccine was produced using the Korean PED epidemic strain KNU-141112. Pregnant sows vaccinated intramuscularly with this monovalent, adjuvanted vaccine at 6 and 3 weeks before farrowing showed an approximately 92% survival rate of their piglets post-challenge with the viral strain. Additionally, there was a significant decrease in the severity of diarrhea and viral shedding in these piglets [54]. Collin et al. created an inactivated vaccine based on the US variant NPL-PEDV 2013 P10.1 strain, which is classified in group G2b [92]. This vaccine, administered intramuscularly, generated a substantial humoral immune response against PEDV, as evidenced by cell-based VN tests. However, the immune response triggered by inactivated vaccines is less comprehensive and less enduring than that induced by LAVs, leading to the need for multiple doses for effective immunization.

### 3.3. Subunit Vaccines

Subunit vaccines offer numerous benefits over whole-virus vaccines, such as enhanced safety, absence of infectious viral RNA, and provision of well-defined, consistent antigens, albeit with lower innate immunogenicity. This lower immunogenicity necessitates the use of adjuvants or immune enhancers for effectiveness [93,94]. The development of a comprehensive subunit vaccine production system is crucial for rapidly addressing sudden epidemic outbreaks.

Researchers have recently utilized various expression systems to focus on the PEDV S protein, a key target in subunit vaccine development due to its multiple biological functions. These include *Escherichia coli* [95,96,97,98], *Bacillus subtilis* [99], Lactobacillus [100,101,102,103], baculovirus [104], yeast [105], HEK293T cells [65,106,107,108] and transgenic plants [109,110] to express either part of the PEDV S protein, such as the COE and S1 regions, or a full-length version. Vaccines developed through these systems and administered to mice or pigs via oral, intramuscular, subcutaneous, or intraperitoneal inoculation have successfully produced high levels of IgG and IgA antibodies. Additionally, the undefined receptor-binding domains (RBDs) and the heptad repeat 1 and 2 segments in the S2 region of PEDV S protein, which represent further potential targets for subunit vaccine development [111,112]. Despite the presence of epitopes in the N and M proteins, there are no reports on subunit vaccines based on these specific PEDV proteins [113,114].

### 3.4. Virus-like Particle Vaccines

VLP vaccines, a subset of subunit vaccines, are created using recombinant protein techniques without the need for a viral replication system [115]. They preserve the native antigenic structure of immunogenic antigens, resembling the structure of wild viruses, but are noninfectious due to the absence of viral nucleic acids [116]. A primary limitation of subunit vaccines is that the proteins expressed are not as immunogenic as the original viral components, necessitating higher antigen quantities for equivalent protection. However, VLP vaccines, which mirror the structure of the actual virus, can elicit stronger antibody responses with fewer antigens than traditional inactivated or subunit vaccines [117]. Moreover, VLPs lack genomic material, ensuring safety without the risk associated with incomplete inactivation of viral replication. The critical advantage of VLP vaccines lies in their capacity to activate B cells and enhance the proliferation of CD4+ T cells and cytotoxic T lymphocytes [118]. The interaction of VLPs with B cells is potent enough to trigger T cell-independent IgM antibody production [119]. Due to their large, unique structures akin to those of microorganisms, VLPs elicit a robust response from the mammalian immune system, attributed to their antigenic array [115]. Studies on the influence of antigen epitope density and arrangement have shown that well-organized antigens can elicit swift responses [120]. On the basis of their dense, repetitive arrays of geometrically organized proteins, VLPs can provoke a robust immune response [115]. In addition to potent B-cell activation, VLPs efficiently stimulate T-cell responses through interactions with antigen-presenting cells, particularly dendritic cells. These cells process VLPs via MHC II molecules, leading to the activation of T helper cells [121]. VLP antigens also enhance dendritic cells’ ability to generate a quicker, more robust, and protective cytotoxic T-lymphocyte response through class I presentation [121].

Coronavirus VLPs are spontaneously generated in cells that express proteins such as M or E, facilitating virus particle assembly [122]. In a study focusing on PEDV VLPs, eukaryotic cells were engineered to express the S, M, and N proteins of the virus, and the resulting immune responses were subsequently observed in mice. Immunization with these PEDV VLPs in mice led to the induction of the production of both IgG and IgA antibodies. These antibodies conferred protection against PEDV infection in Vero cells. Using a multicistronic baculovirus expression system, Hsu and colleagues successfully created VLPs incorporating the S, M, and E structural proteins of PEDV [123]. Their research revealed that pigs immunized intramuscularly developed a systemic immune response characterized by anti-PEDV S-specific IgG and cellular immunity. When exposed to the virus, these pigs demonstrated a degree of resistance superior to that of the control pigs, although this resistance did not extend to preventing viral excretion in feces. Additionally, the joint administration of the mucosal adjuvants CCL25 and CCL28 resulted in a modest increase in IgG production and fecal score. However, this treatment did not significantly impact the induction of the production of neutralizing antibodies or result in a reduction in virus shedding.

Researchers have developed a VLP-based vaccine that integrates the B-cell epitope 748YSNIGVCK755 from the S protein of PEDV into the hepatitis B virus core capsid (HBcAg) [98,124]. This vaccine has been tested in both mouse and pig models. In the mouse model, intraperitoneal injection of the vaccine-induced increases in the serum IgG and IgA concentrations, although it did not stimulate the production of fecal IgA. Notably, the vaccine elicited a substantially greater virus neutralization response in gilt milk, contributing to the mitigation of clinical symptoms in piglets experimentally infected with PEDV. Piglets born to vaccinated gilts exhibited accelerated recovery from the disease, reduced damage to the small intestine, and a greater survival rate 10 days post-challenge.

ADDomer, a nanoparticle scaffold derived from adenovirus, is built on multimeric proteins, allowing for easy incorporation of various immunogenic epitopes from pathogens [125]. In preclinical studies, an ADDomer-based vaccine harboring the chikungunya virus E2 protein exhibited notable immunogenicity [125]. Additionally, an ADDomer VLP expressing the RBD of the SARS-CoV-2 S protein elicited a strong humoral immune response and the production of neutralizing antibodies in mice [126]. In piglet immunization experiments, a recombinant ADDomer-VLP vaccine was tested; this vaccine expressed the SS2 and 2C10 regions of the PEDV S protein and the A and D sites of the TGEV S protein [127]. This recombinant ADDomer-VLP vaccine exhibited robust immunogenicity, successfully inducing the production of neutralizing antibodies against both PEDV and TGEV. Furthermore, it led to increased levels of IFN-γ, IL-2, and IL-4 in piglets and enhanced cytotoxic T-lymphocyte activity in their peripheral blood.

### 3.5. Viral Vector Vaccines

Focusing on the PEDV S protein, researchers have developed viral vector vaccines using various expression systems, including adenovirus, vesicular stomatitis virus, and poxvirus.

Recent studies have increasingly indicated that the use of vaccines based on recombinant adenovirus live vectors is an emerging and effective strategy for preventing viral infections. Liu et al. developed a recombinant adenovirus that expresses the S protein of PEDV, termed rAd-PEDV-S. This vaccine candidate was tested in 4-week-old pigs, which revealed that it induced a notable PEDV-specific immune response [128]. Moreover, rAd-PEDV-S offered protection to pigs challenged with a highly virulent strain of PEDV. Do et al. developed a novel recombinant adenovirus vaccine, rAd-LTB-COE, which carries the genes for heat-labile enterotoxin B (LTB) and the core neutralizing epitope (COE) of PEDV [129]. Administering this vaccine through three doses, either intramuscularly or orally, at two-week intervals successfully stimulated strong humoral and mucosal immune responses. Additionally, it was found to enhance the cell-mediated immune response in immunized mice. Notably, the neutralizing antibodies produced were effective against both the vaccine strain of PEDV and newly emerging strains of the virus. Tests conducted in piglets demonstrated that rAd-LTB-COE effectively elicited robust immune responses, underscoring its potential as an immunization agent. Four innovative human adenovirus 5-based vaccines, each lacking the ability to replicate, were designed to express the S and/or S1 glycoproteins of the GIIa and GIIb strains (rAD5-PEDV-S) [130,131]. The effectiveness of these vaccines was tested in vivo. In particular, the rAd5-PEDV-S vaccine induced a significant PEDV-specific humoral immune response in sows, which included IgA and IgG in colostrum and serum, as well as circulating neutralizing antibodies. The performance of this vaccine surpassed that of the commercial inactivated vaccine, with intramuscular administration proving more effective than intranasal delivery. After exposure to PEDV, five-day-old piglets born to sows vaccinated with rAd5-PEDV-S via the intramuscular route experienced ameliorated diarrhea and weight loss. These piglets also had significantly lower fecal PEDV RNA expression than did the other piglets in the other groups. Notably, all piglets in the rAd5-PEDV-S intramuscular group survived the infection period.

Ke et al. employed a greatly weakened recombinant vesicular stomatitis virus (rVSVMT) as a vector for expressing the PEDV S protein, generating VSVMT-SΔ19 [132]. This vector successfully stimulated PEDV-specific immunity in pigs when administered via intramuscular injection, although it was not effective when given intranasally. When sows were immunized with VSVMT-SΔ19, protective lactogenic immunity was conferred to their piglets against a potent G2b PEDV challenge.

Yuan et al. developed a novel live recombinant vaccine using a swinepox virus vector to express a truncated S protein (rSPV-St) from the recent PEDV strain SQ2014 and assessed its immunogenicity and effectiveness in pigs [133]. The administration of this vaccine to swine resulted in a strong antibody response specific to the homologous PEDV SQ2014 strain. Notably, the serum IgA concentrations in the animals vaccinated with rSPV-St were significantly greater than those in the pigs given inactivated vaccines. The efficacy of the antibodies produced upon immunization with the rSPV-St vaccine in preventing PEDV infection was evaluated using a passive-transfer model. Piglets challenged with the homologous virus SQ2014 were completely protected by sera from pigs that underwent rSPV-St vaccination, but the same was not true when the heterologous strain CV777 was used as the challenge agent. This lack of protection against a heterologous virus might be attributed to amino acid sequence variations in the S proteins of the two viruses, particularly in neutralizing epitopes.

Additionally, a new recombinant Orf virus (ORFV) expressing the full-length S protein of PEDV was created, and its immunogenicity and protective efficacy were tested in pigs [134]. Three intramuscular immunizations with recombinant ORFV-PEDV-S in 3-week-old pigs induced substantial serum IgG, IgA, and neutralizing antibody responses against PEDV. Moreover, intramuscular immunization with the recombinant ORFV-PEDV-S virus protected pigs from the clinical symptoms of PED and diminished virus shedding in feces following the challenge.

These findings suggest that various viral vectors could be carriers of a PEDV vaccine, indicating that these vectors could be viable options for the future prevention and management of PED. However, their effectiveness requires further enhancement in future studies.

### 3.6. Nucleic Acid Vaccines

Nucleic acid vaccines, such as DNA and mRNA vaccines, offer several advantages: they are cost-effective and safe, can be rapidly produced, and are relatively straightforward to design [135,136]. A key benefit is the universality of the established platforms. Once established, these platforms require only the synthesis and insertion of the core antigen gene into a suitable expression vector, potentially accelerating the development of vaccines for new pandemics [137]. However, there are challenges. For instance, bivalent DNA vaccines combining PEDV with either porcine rotavirus or TGEV have been created but not yet tested in clinical trials due to limited immunogenicity [138,139,140]. Enhancements in DNA vaccines have been achieved through various techniques, such as codon optimization, careful promoter selection, plasmid vector backbone optimization, and molecular adjuvant addition. However, significant advancements are still needed for DNA vaccines to become more effective at preventing diseases.

RNA vaccines show considerable promise. Notably, as early as 2014, Harrisvaccines in the United States developed a PEDV mRNA vaccine using SirraVaxSM technology, which was conditionally licensed in the United States; however, mRNA vaccines, while more effective immunologically than DNA vaccines, tend to be less stable in the body [136]. Self-amplifying RNA (saRNA) vaccines, which are developed via the incorporation of alphavirus replicase and viral genes, replicate RNA in the cytoplasm [141,142]. These vaccines have shown promising results; for example, a saRNA vaccine based on influenza hemagglutinin outperformed an mRNA vaccine in mice [143]. Additionally, a saRNA vaccine encoding the SARS-CoV-2 S protein and encapsulated in lipid nanoparticles showed robust neutralizing effects in mice, as evidenced by the high antibody titers [144]. Thus, developing saRNA vaccines and delivery systems might be key to overcoming current limitations in creating effective mRNA vaccines for PEDV and other coronaviruses.

## 4. Conclusions

Despite numerous efforts to create effective vaccines, the majority of these agents have not succeeded in triggering mucosal immunity, as evidenced by studies [65,145]. While the immunogenicity of live attenuated and inactivated vaccines is generally considered more potent than that of other vaccines [146], LAVs are associated with safety concerns, including the risk of genetic recombination or reversion to virulence via recombination with wild-type strains. Furthermore, both vaccine approaches necessitate considerable time for development [147].

In the quest for innovative solutions, next-generation vaccines are expected to swiftly yield vaccines, addressing the high mutation rates of RNA viruses [22]. The subunit vaccine strategy is considered safer, yet it falls short of achieving optimal immunogenicity. With advancements in biotechnology, employing VLPs as advanced subunit vaccines has emerged as a viable option. This approach not only reduces development time and ensures safety but also achieves adequate immunogenicity. Thus, this approach represents a novel, balanced method for vaccine development, achieving an optimal balance between safety and immunogenicity.

## Figures and Tables

**Table 1 vaccines-12-00212-t001:** Vaccine strategies for PEDV.

Vaccine Platform	Antigen	Development	Advantages	Limitations
Live-attenuated virus	Whole virus	Multiple passages in cell culture Reverse genetics	High immunogenicity (strong cellular and humoral immune responses); can be given orally and in a single dose	Low safety (reversion to virulence, recombination with a field strain)
Inactivated virus	Whole virus	Inactivation with chemicals	Easy to prepare; high safety	Needs adjuvant; multiple injections; low immunogenicity (Th-2 skewed immune response)
Subunit	Spike protein	Antigen expressed in mammalian, baculovirus, yeast, or plant cells	Enhanced safety; rapidly produced	Expensive; needs adjuvant; limited immunogenicity
Viral vector	Spike protein	Adenovirus, vesicular stomatitis virus, poxvirus engineered to express spike protein	Strong cellular and humoral immune responses; intrinsic adjuvant properties; can be given orally	Pre-existing immunity against viral vector
Nucleic acid	Spike protein	Genes (mRNA or DNA) encoding spike proteins	Cost-effective; high safety; rapidly produced; easy to prepare	Limited immunogenicity (Th-1 skewed immune response)
Virus-like particle	Spike and other structural (envelope, membrane, or nucleocapsid) proteins		Strong immune responses; activates both B cell and T cells; high safety	High cost; low yield

## Data Availability

Not applicable.

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
