# Peer review of "Porcine Epidemic Diarrhea: Insights and Progress on Vaccines"

_vaccines, 2024, doi:10.3390/vaccines12020212_

Round 1
Reviewer 1 Report
Comments and Suggestions for Authors
This review is an excellent summary of the current knowledge of vaccines not only for PED but also for other coronavirus infection. A few minor revisions are listed below.
1. Line 27. Delete since?
2. Lines 35-36. Alphacoronavirus, Coronaviridae, and Nidovirales should be in Italic.
3. Line 45. Delete virus.
4. Line 54. 2013–2014 epidemic should read 2013–2014 pandemic (line 33) or vice versa_
5. Lines 76-78. This sentence should be revised.
6. Lines 106 and 224. PED not PEDV.
7. Line 124. LAV not live vaccine.
8. Line 158. IFN should be spelled out.
9. Line 228. Delete fecal.
10. Line 231. VN not viral neutralization.
11. Line 250. HR should be spelled out.
12. Lines 274-275. This sentence should be modified.
13. Lines 279 and 282. PEDV not PED.
14. Line 383. RV should be spelled out.
Comments on the Quality of English Language
Minor editing of English language required.
Author Response
Thank you for your comments. We have revised the manuscript on all points according to your comments.
Reviewer 2 Report
Comments and Suggestions for Authors
This review by Jung-Eun Park is well-written. It summarized the current research on the development of different types of PEDV vaccines. It provides valuable information for readers in the field. My only suggestion is that to have a table listing the vaccines mentioned in the study and summarize the key information of each vaccine.
Author Response
Thank you for your comment. We have added a table listing the vaccines mentioned in the review and summarizing key information about each vaccine. Please check Table 1.
Reviewer 3 Report
Comments and Suggestions for Authors
Author summarized exiting vaccine platforms for PED very well. However, a detailed perspective on how the field will progress is lacking.
Comments on the Quality of English LanguageNo big issues identified.
Author Response
Thank you for your comment. We have added our perspective on next-generation PED vaccines in conclusion (lines 411-418).
Reviewer 4 Report
Comments and Suggestions for Authors
The review is well-written and properly referenced, providing a comprehensive overview of the current vaccines against PEDV, ongoing developments and challenges.
The revised manuscript was in the "review" format, and it discussed the different vaccines that are available and in development against PEDV. It was not a research study, but rather a review of the literature on the subject. Therefore, many of the questions that follow are not applicable, as there was no starting hypothesis, methodology, results, controls, or the possibility of comparison with previously published studies.
The subject of the manuscript is highly relevant, as PEDV epidemics and outbreaks are occurring worldwide and pose a threat to the swine industry. Although vaccines have been developed against the virus, the continued emergence of new variants, particularly those resulting from recombination, has rendered some of these vaccines ineffective. This presents significant challenges to the prevention and control of PEDV. The manuscript aims to provide an overview of the current situation and vaccination efforts to eradicate PEDV.
The authors have done an excellent job of providing information about the virus and the history of epidemics to help readers understand the issue. This section is informative without being too lengthy. They have also given an update on various vaccination strategies that have been attempted in the past, followed by a comprehensive review of different types of vaccines under development, including their advantages and challenges. The author has used 147 highly relevant references to document the review comprehensively.
This manuscript provides an up-to-date report on the progress of PEDV vaccine development, which has not been covered in recent publications.
Please change the word "chapter" to "review" on line 110 and check reference 148 (line 215); it's missing from the reference section.
Author Response
Thank you for your comment. We have revised the manuscript on all points according to your comments. I am sorry but reference 148 is a typo.